# communications
# engineering

# Electroacoustic tomography for real-time visualization of electrical field dynamics in deep tissue during electroporation

Yifei Xu [1], Leshan Sun[1], Siqi Wang[1], Yuchen Yan[1], Prabodh Pandey[2], Vitalij Novickij [3,4✉] & Liangzhong Xiang [1,2,5✉]

Despite the widespread applications of electroporation in biotechnology and medicine, monitoring the distribution of deep tissue electrical fields in real-time during treatment continues to pose a challenge. Current medical imaging modalities are unable to monitor electroporation during pulse delivery. Here we propose a method to use electroacoustic tomography (EAT) to prompt the emission of broadband ultrasound waves via electrical energy deposition. EAT boasts submillimeter resolution at depths reaching 7.5 centimeters and can deliver imaging speeds up to 100 frames per second when paired with an ultrasound array system. We've successfully detected EAT signals at electric field strengths ranging from 60 volts per centimeter to several tens of kilovolts per centimeter. This establishes EAT as a potential label-free, high-resolution approach for real-time evaluation of deep tissue electroporation during therapeutic procedures.

[1] The Department of Biomedical Engineering, University of California, Irvine, CA, USA. [2] The Department of Radiological Sciences, University of California at Irvine, Irvine, CA, USA. [3] Institute of High Magnetic Fields, Vilnius Gediminas Technical University, Vilnius, Lithuania. [4] Department of Immunology, State Research Institute Centre for Innovative Medicine, Santariškių 5, 08410 Vilnius, Lithuania. [5] Beckman Laser Institute & Medical Clinic, University of California, Irvine, Irvine, CA, USA. ✉email: vitalij.novickij@vilniustech.lt; liangzhx@hs.uci.edu

The application of electricity in humans can be traced back to the 18th century when electric fields were first observed to cause tissue damage[1–5]. In the decades that followed, researchers investigated the effects of electric fields on biological systems, leading to the discovery of electroporation in 1982 by Neumann et al.[1]. This technique involves the application of pulsed electric fields that increase the permeability of the cell membrane, allowing for the uptake of foreign molecules such as drugs or DNA. Since its discovery, electroporation has been extensively used in various applications, including DNA transfection[6,7], electrochemotherapy[8,9], tissue ablation[10,11], compound extraction[12], microbial inactivation[13] and more. Electroporation-based therapies offer new ways of delivering therapeutic agents into cells, with the efficacy of these treatments depending on the parameters of the electrical pulses (including amplitude, pulse duration, the number and the repetition rate of pulses)[14]. Currently, the field is dominated by microsecond range protocols, however, a tendency towards the development of shorter pulse (i.e. nanosecond pulses) protocols is observed. The advent of a high-intensity, nanosecond pulsed electric field (nsPEF) enabled a range of intracellular studies and applications[15]. nsPEF has been demonstrated to be effective in several applications, including sterilization, wound healing, activation of neurons and myocytes, cell proliferation, modulation of gene expression, and novel cancer treatment[16]. Specifically, nsPEF has been shown to induce apoptosis in cancer cells by disrupting the membrane integrity and intracellular structures, leading to cell death[17]. Finally, the application of nsPEF allows better Joule heating and oxidative damage management[18], improves spatial electric field distribution due to higher frequency component[19,20], results in reduced muscle contractions and potentially less pain[21]. Nevertheless, nsPEF pulses are more challenging to generate requiring the latest semiconductor technologies, in many cases impedance matching must be performed to prevent pulse form alterations and more importantly the complexity of metrology and characterization of the treatment is increased. To improve the efficacy and repeatability, numerical modeling can be used for treatment planning[22–25], and medical imaging modalities can be used for post-treatment evaluation[26–29]. However, current imaging methods are unable to monitor the electroporation process in real time during pulse delivery. CT imaging is commonly used to guide the placement of electrodes during IRE ablation for liver cancer treatment, but it requires the patient to remain still during the procedure to prevent motion artifacts caused by the patient's breath and can expose them to higher radiation levels than ultrasound-guided IRE. Both CT and MRI imaging can be used to assess the efficacy of IRE post-treatment, and contrast agents may be used in some cases to enhance visualization in cases where residual viable tumor tissue needs to be identified and further treatment is required[30]. Electrical impedance tomography (EIT) has been proposed as a solution, but it faces technical challenges resulting in poor spatial resolution[31]. Magnetic resonance electrical impedance tomography (MREIT) has been developed for high-resolution conductivity images, but it requires an expensive MRI scanner and has a poor temporal resolution[32,33]. Further advancements are needed to overcome these limitations for the practical implementation of MREIT.

Here, we investigated electroacoustic tomography (EAT) as a new imaging method to address challenges in monitoring electroporation. EAT provides high-resolution, label-free, non-invasive measurements of electrical field distribution up to several centimeters deep in tissue. Specifically, we explored the use of nanosecond Pulsed Electric Field (nsPEF) excitation with broadband ultrasound detection[34,35]. Our novel EAT imaging system with 128 ultrasound elements and a 128 parallel data acquisition system allows for real time, high-resolution tomographic imaging. We demonstrated the ability to visualize the electrical field distribution in soft tissue and monitor electroporation in real time, indicating the great potential of EAT in clinical translation.

## Results

**Principle of EAT imaging.** Electroacoustic tomography detects electric fields in soft tissues by inducing an acoustic pressure with a high voltage pulse. If the electric pulse is short enough to satisfy both thermal and stress confinements, the thermal diffusion and stress propagation of tissue during pulses delivery can be negligible. Thus, the initial acoustic pressure $p_0$ is proportional to the local energy deposition and can be expressed as follows[36]:

$$p_0(r,t) = \frac{\beta(r)}{\kappa(r)\rho(r,t)C_v(r)}H(r,t) \qquad (1)$$

where $\beta(r)$ is the thermal coefficient ($K^{-1}$), $\kappa(r)$ denotes the isothermal compressibility ($Pa^{-1}$), $\rho(r,t)$ is the mass density ($g \cdot m^{-3}$), $Cv(r)$ is the specific heat capacity at constant volume ($J \cdot g^{-1} \cdot K^{-1}$), and $H(r,t)$ denotes the deposited electrical power density ($J \cdot m^{-3}$) in tissue. The electroacoustic wave's amplitude correlates to the energy deposited, allowing for the reconstruction of electric field distribution. Joule's and Ohm's laws determine the relationship between electrical power density and electric field, and this allows for expressing the initial pressure rise in the target as[35].

$$p_0(r,t) = \frac{\beta(r)\sigma(r)}{\kappa(r)\rho(r,t)C_v(r)}E(r)^2 g(t) \qquad (2)$$

where $E(r)$ is the electric field ($V \cdot m^{-1}$) at the position of $r$ and $g(t)$ denotes the voltage pulse width (s). $\sigma(r)$ denotes the specific electrical conductivity ($S \cdot m^{-1}$), while it is not a constant. It has been shown that the electrical conductivity of the tissue increases irreversibly during electroporation and that this increase is related to the amplitude and number of pulses[37]. It has also been shown that there is a limit to the effectiveness of the number of pulses, and that conductivity will not increase further when the number of pulses is above a certain threshold[38]. In this study, a high number of pulses was used, so the conductivity $\sigma(r)$ was simplified to a constant.

The acoustic signal can be induced because of the thermoelastic expansions of the tissue. The electrical energy will be absorbed by tissue and transformed into heat. A heating function for electroacoustic wave generation is given as follows[36]:

$$H(r,t) = \eta_E A_e(r,t) \qquad (3)$$

Where $\eta_{th}$ is the electrical energy absorption, and $A_e$ donate the electrical energy deposition. Thus, the wave equation used to demonstrate the EA signal generation process can be modeled as

$$\left(\nabla^2 - \frac{1}{v_s^2}\frac{\partial^2}{\partial t^2}\right)p(r,t) = -\frac{\beta\eta_{th}}{C_p}\frac{\partial H(r,t)}{\partial t} \qquad (4)$$

where $v_s$ is the speed of sound, $p(r,t)$ is the electroacoustic pressure at position $r$ and time $t$, and this wave equation can be solved by Green's function and simplified to[35]:

$$p(r,t) = \frac{1}{4\pi v_s^2}\int dr' \frac{1}{|r-r'|}p_0(r')\delta\left(t - \frac{|r-r'|}{v_s}\right) \qquad (5)$$

where the $p_0(r')$ represent the initial electroacoustic pressure rise. And universal back-projection (UBP) method is employed to reconstruct EAT image in this study[39]. Therefore, the EAT imaging will reveal: 1) the map of the electrical field energy deposition during the electroporation deliveryn and 2) the dose amount deposited to the target volume. These information will be very important to determine the effectiveness of electroporation.

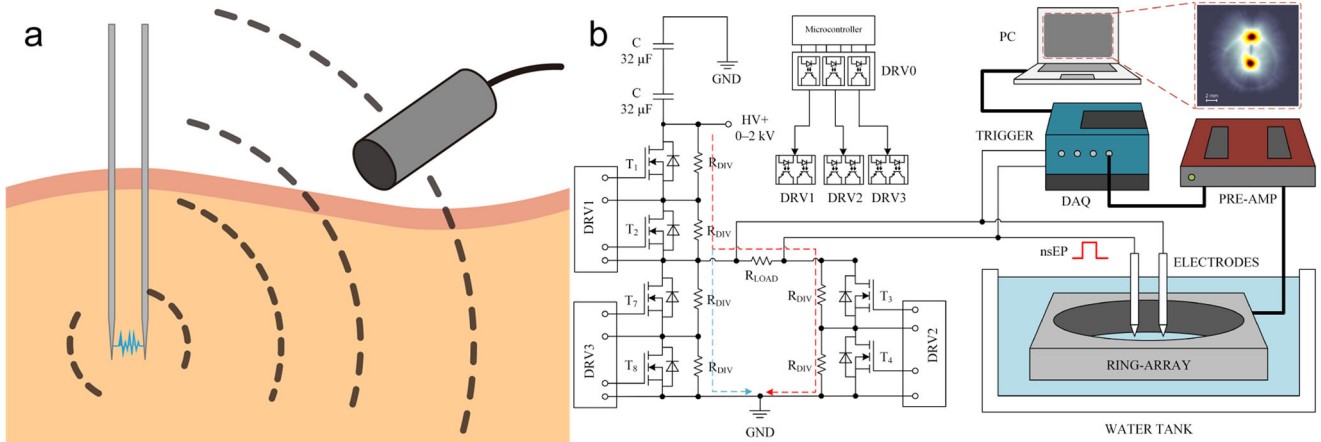

**Fig. 1 Illustration of the principle of electroacoustic tomographic (EAT) imaging and a typical system. a** The EAT imaging principle is shown schematically, where electrical energy deposition stimulates the emission of broadband ultrasound waves, enabling submillimeter resolutions over centimeters of tissue depth in real time. **b** The EAT imaging device comprises a 100 nanoseconds Pulsed Electric Field (nsPEF) with a repetition frequency of 1 MHz and an amplitude of 2000V for electroporation, combined with a broad-bandwidth ultrasound ring array (128 elements) detection system. PC: personal computer; TRIG: trigger signal; FG: function generator; nsEP: nanoseconds electrical pulse; PRE-AMP: pre-amplifier; DAQ: data acquisition system.

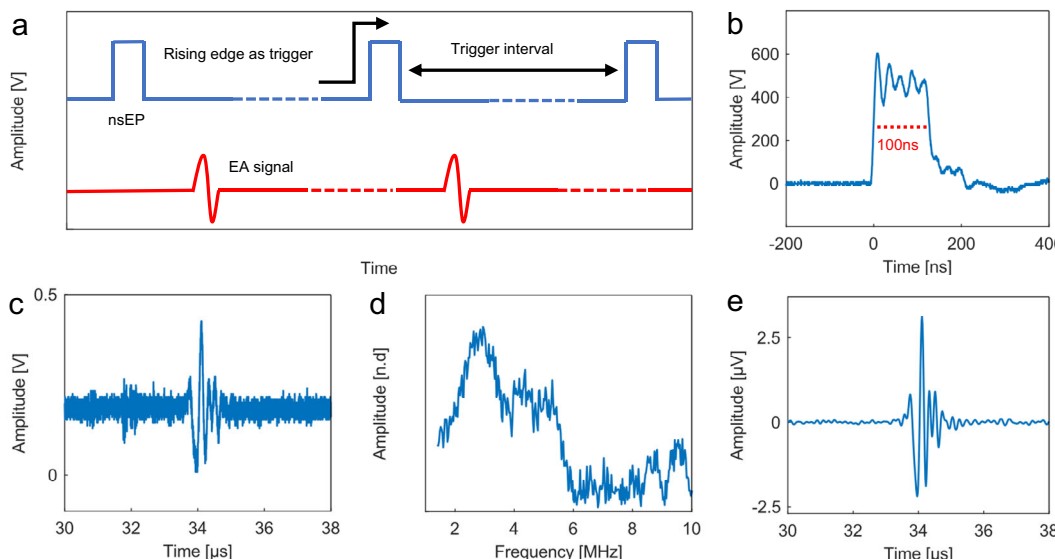

**Fig. 2 Electroacoustic signal detection and characterization. a** A periodic rectangular waveform with peak-to-peak voltages of 500 Vpp (blue), repeating at 1000 Hz for nanosecond pulsed electric field electroporation and generate electroacoustic signals (red). **b** The electric pulse profile has a pulse width of ~100 ns and a rising time of ~15 ns. **c** A typical 5 MHz ultrasound transducer detects the electroacoustic signal. **d** The frequency spectrum of the electroacoustic signal shows most frequency components are within 2-6 MHz. **e** Filtering the electroacoustic signal with 6 MHz low pass filter results in an SNR of 28.5 dB. nsEP: nanosecond electrical pulse.

Figure 1a illustrates the EAT imaging schematic diagram, which consists of nanosecond electrical pulses to generate detectable electroacoustic signals. We designed a high-voltage nanosecond pulse generator (VilniusTECH, Vilnius, Lithuania) and a tungsten electrode pair to deliver electrical energy to the tissue. This excitation source can produce a nearly square pulse with an FWHM of 100 ns and a repetition rate of up to 1 MHz (shown in Fig. 2a). Electroacoustic signals were collected using a 5 MHz ultrasonic transducer (A309S-SU, Olympus, USA), amplified with a low-noise preamplifier (62 dB, Photosound, USA), and digitized with an oscilloscope (DSOX2024A, Keysight Technologies, USA). For the rest of the experiments, as shown in Fig. 1b, a ring ultrasound array with 128 elements and a center frequency of 5 MHz was used (PA probe, Doppler Ltd., Guangzhou, China). Raw ultrasound data were received by a 128-channel data acquisition (SonixDAQ, Ultrasonics, BC, Canada) and stored for processing. No mechanical scanning is required, and the imaging speed is limited only by the pulse repetition rate and the limited flight time of the ultrasound signal. The signal was reconstructed using a filtered back-projection algorithm[39], providing real-time observation of electric field dynamics in the tissue during electroporation.

**Electroacoustic signal detection and characterization.** To characterize the electroacoustic signal, a 5 MHz commercial ultrasound transducer is used to detect the EA signal (A309S-SU, Olympus, USA). A custom pulse generator is used to produces a square wave pulse with an amplitude of 500 V, a pulse width of 100 ns, and a rising edge of about 15 ns (Fig. 2a-red and Fig. 2b).

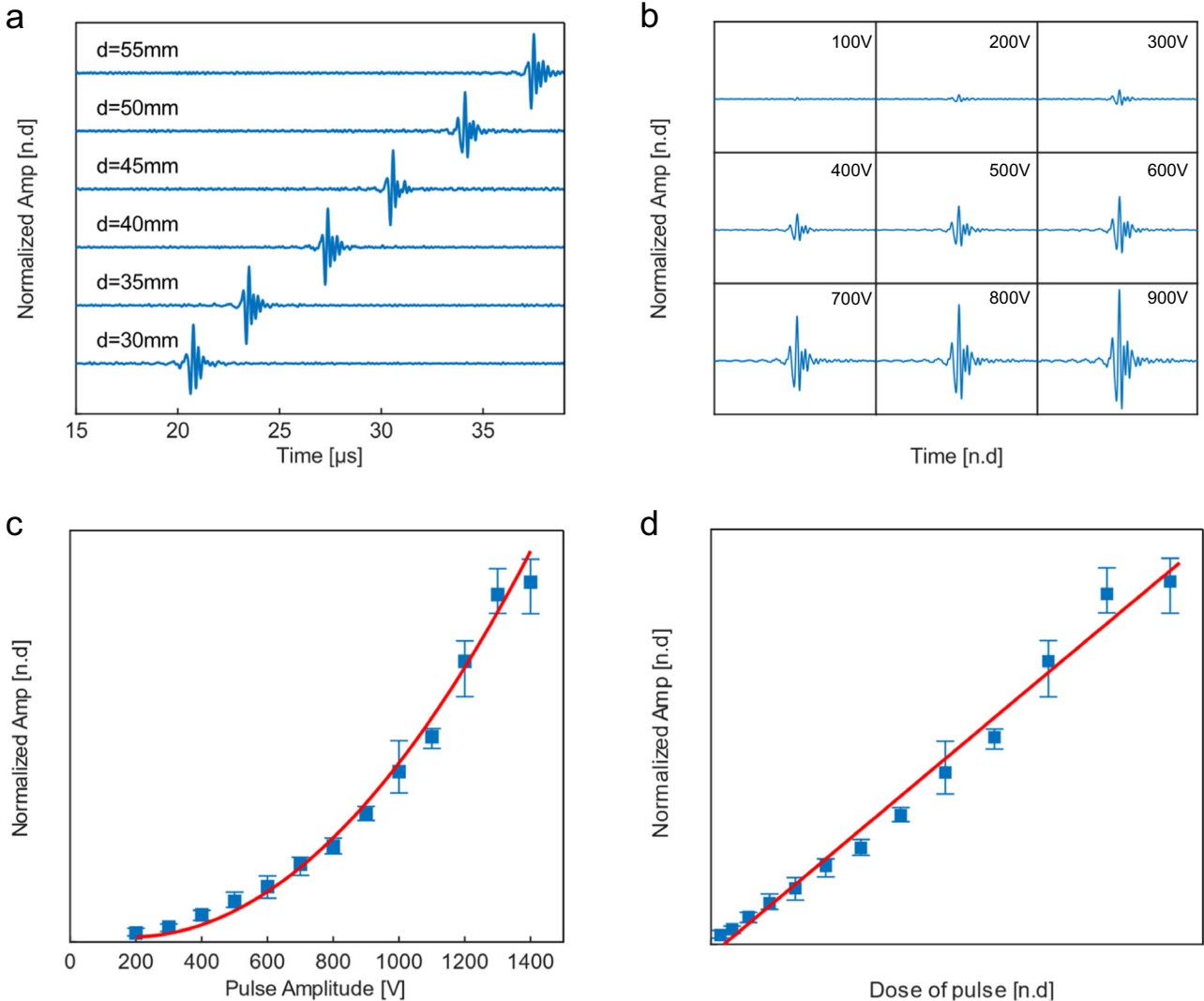

**Fig. 3 Correlation between electroacoustic signal and electroporation. a** Displays representative detections of acoustic signal responses to ultrasound transducer locations ranging from 30 mm to 55 mm, corresponding to electroacoustic signal time-of-flight information from 20.8 μs to 37.5 μs. **b** The electroacoustic signal strength increases with input voltage, ranging from 100 volts to 900 volts. **c** The relationship between electroacoustic signal strength and input voltage follows an exponential function, which is associated with electrical energy deposition and the electroporation effect. **d** The energy delivered by the electrical pulse (Dose) is linearly related to the amplitude of the electroacoustic signal. The error bar (blue, minimum and maximum values in a dataset, $N = 5$), and a curve fit of EA signal amplitude (red).

Electroacoustic signal (Fig. 2a-blue and Fig. 2c) with a dominant frequency of 2-6 MHz (Fig. 2d) is generated when the pulsed electric field is delivered. A low-pass filter is applied to remove high-frequency noise, resulting in a clear signal with high SNR (28.5 dB) (Fig. 2e). These results suggest that the electroacoustic signal falls within the medical ultrasound range and can potentially be detected with a clinical ultrasound imaging probe.

**Correlation between electroacoustic signal and electroporation.** We conducted experiments to establish the relationship between the intensity of the electroacoustic signal and input voltage to demonstrate its application for monitoring electroporation. Figure 3a displays the linear relationship between the time delay of the ultrasound signal generated and the distance of the electrodes from the ultrasound transducer, which was scanned in steps of 5 mm. The induced wave's velocity was determined to be 1497 m·s$^{-1}$, which is close to the speed of sound in saline ($c_s = 1507$ m·s$^{-1}$) at 25 °C[40], indicating that the waves were ultrasonic waves which we called electroacoustic waves.

We also investigated how the electrical parameters affect the electroacoustic signal intensity by recording the electroacoustic signal at a fixed distance of 50 mm from the ultrasound transducer while varying the electric field strength between 0.4-2.8 kV·cm$^{-1}$ for 100 ns pulses (Fig. 3b). The sensitivity of the system was found to be 0.06 kV·cm$^{-1}$ (Supplementary Fig. S2, Supplementary Note. S2). The electroacoustic signal amplitude was found to be exponentially dependent on the input voltage (Fig. 3c), and linear with respect to input electrical energy (Fig. 3d), indicating that electroacoustic measurements can quantitatively monitor nsEP during electroporation. In addition to the electrical pulses, the electrical parameters of the phantom also have an impact on the electroacoustic signal. We found that the intensity of the electroacoustic signal increases linearly with the ion concentration of the sample (Supplementary Fig. S3, Supplementary Note. S3).

**EAT imaging of electrical field in soft tissue.** To test the hypothesis that EAT imaging can visualize the electrical field

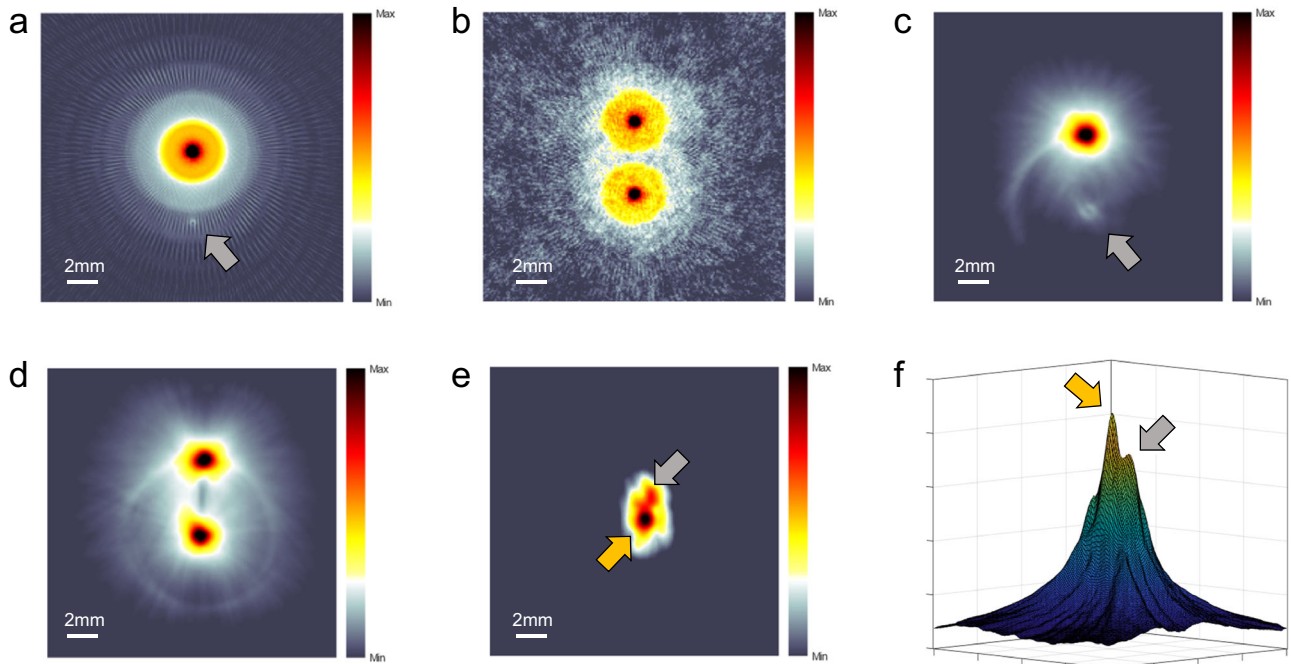

**Fig. 4 EAT imaging of electrical field distribution both in water and soft tissue.** Electric field distributions were simulated in COMSOL with one positive electrode paired with one ground electrode in (**a**), and with two positive electrodes in (**b**). The EAT images from the experiments in (**c**) and (**d**), respectively, correlate well with the simulated results. The EAT map in (**e**) shows the electrical field distribution during electroporation in soft tissue, while the 3D intensity profile of the electrical field distribution is displayed in (**f**). Orange arrow: positive electrode; Gray arrow: ground electrode.

distribution during electroporation, we conducted simulations and experiments in water and soft tissue using one positive and one ground electrode (1p1g) and two positive electrodes (2p), respectively. In the simulation, the electric field and energy deposition distribution were calculated using COMSOL and then imported into the Matlab k-wave toolbox to simulate EA wave propagation and reconstruction of EAT images using a two-dimensional time-reversal image reconstruction algorithm. Simulation results in Fig. 4a and b show that the energy of the electric field is concentrated around the positive electrode in the 1p1g electrode pair, while it is similar around both electrodes in the 2p electrode pair. The energy of the electric field decreases with distance, leading to a weaker signal intensity in the reconstructed image.

Figure 4c, d present reconstructed EAT images of a 1% concentration saline agar phantom with 1200 V amplitude ($2.4\,kV\cdot cm^{-1}$ electric field intensity) and 100 ns pulse width. A previous study has successfully conducted nanosecond electroporation on humans using ultra-high voltages up to 30 kV, and its safety has been established. Therefore, this experiment was conducted well within the established safety threshold. The reconstructed images reveal an electric field distribution similar to the simulation results, with electroacoustic signals visible around both electrodes. A ring artifact around the lower electrode is due to inconsistent vertical positioning of the electrodes, a common issue with self-made electrodes. Figure 4e and f show *ex-vivo* experiments with fresh chicken breasts using a 1p1g tungsten electrode pair with a spacing of 1 mm and 1000 V amplitude ($10\,kV\cdot cm^{-1}$ electric field intensity). The higher-intensity electric fields around the positive and ground electrodes are less regular due to muscle fiber orientation, with the peak map in Fig. 4f clearly identifying the positions of the electrodes. These results demonstrate EAT imaging's ability to detect electrode locations and surrounding electric field distribution in soft tissue.

**Real-time visualizing electrical field dynamics**. Figure 5 showcases EAT imaging's ability to spatially resolve electric field

dynamics as the input voltage increases. The experimental setup is illustrated in Fig. 1b, which uses a 127 μm diameter electrode pair with a reduced distance between electrodes to achieve a stronger pulsed electric field intensity. The EAT device captures the spatiotemporal dynamics of the local electric fields in the chicken breast using a ring array ultrasound transducer. The amplitude of the pulse is gradually increased from 100 V to 1000 V in three seconds with a repetition rate of 1 kHz, while the high-speed data acquisition system acquires ultrasound signals in 128 channels at a sampling rate of $40\,MHz\cdot s^{-1}$. The captured data contains 3000 raw frames with 1 ms intervals, and every 10 adjacent frames are signal-averaged to improve the signal-to-noise ratio, resulting in a 3-second EAT video of 100 frames per second (Supplementary Movie S1). The spatiotemporal dynamics of the electric field are captured at a temporal resolution of 10 ms. The spatial resolution of 138um has been demonstrated in this system[41]. In Fig. 5, eight frames of the video are displayed, with the first frame starting at t = 375 ms and each frame interval being 375 ms. The reconstructed images show an increasing extent of the EA signal as the EA signal continues to increase from frames 3 to 8. The strongest signals are distributed around the electrodes, with a clear transition region from the electrodes to the edges, corresponding to the dynamic trend of the electric field induced by the enhancing pulse. The first frame only detects the background signal due to the weak electroacoustic signal. In the second frame, the EA signal begins to emerge from the background, although the signal is still weak due to the 300 V pulse amplitude, the position of the electrodes remains discernible.

Figure 6 shows the spatial dynamics of the EAT image with a fixed pulse amplitude of 1000 V, resulting in a pulsed electric field of $10\,kV\cdot cm^{-1}$. The electrode and chicken breast were moved within 3 seconds, and the image capture protocol was the same as in Fig. 5. The resulting video contained 300 frames, with frames 1-8 showing the electrodes moving from the edge to the center of the ring array (Supplementary Movie S2). As the electrodes

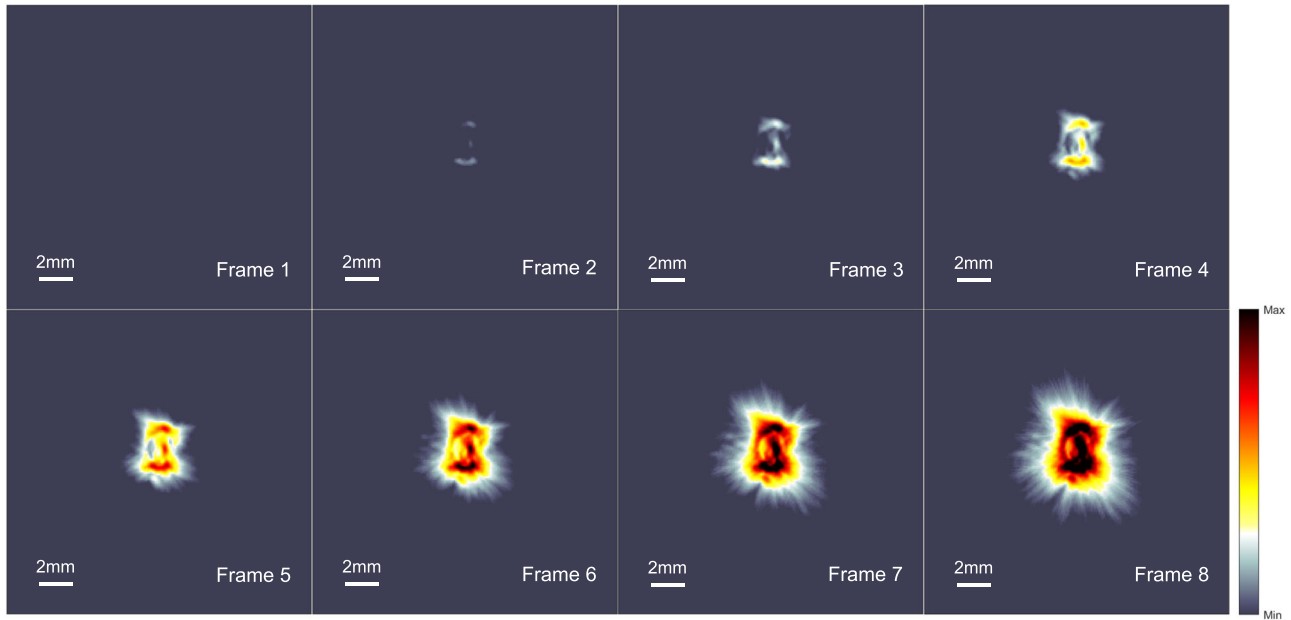

**Fig. 5 Real-time visualizing electrical field dynamics with different voltages.** The electroacoustic signal dynamics were reconstructed while incrementally increasing the pulse intensity from 100 V to 1100 V on the chicken breast sample. Frames 1-8 show the changes in the electroacoustic tomography images over a 3-second period. The first frame, at t = 375 ms, shows no valid signal captured. The interval between frames is 375 ms, revealing an expanding range of the electric field. The image intensity was normalized for comparison purposes. (see the video in the Supplementary Movie S1).

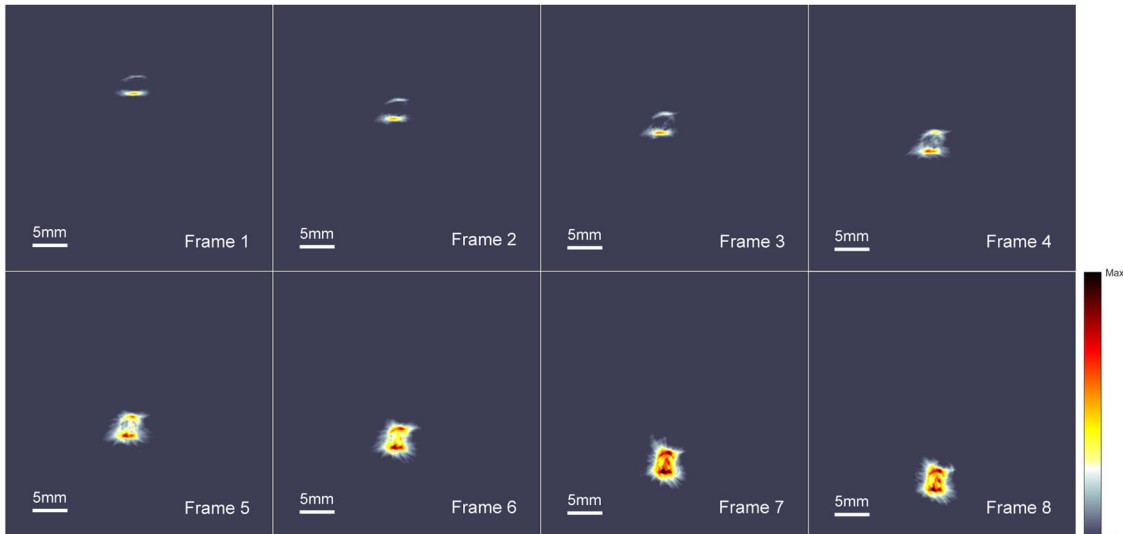

**Fig. 6 Real-time visualizing electrical field dynamics with different locations.** Frames 1-8 illustrate the variations in the electrical field as the position of the electrodes is changed and the interval between each frame is 375 ms. The position of the electrodes moves from the edge to the center of the annular ultrasound array transducer, during which the EA signal is captured by a more effective view of the transducer unit. As a result, increasingly large regions of strong signals can be observed in the EAT images reconstructed by the back-projection algorithm. The viewing window is consistent across all frames, and the signal intensity is normalized for comparison. (see the video in the Supplementary Movie S2).

moved closer to the center of the array, the signal intensity and range gradually increased due to the back-projection reconstruction algorithm used for the ring array. This demonstrates the ability of EAT imaging to capture electric field dynamics over a wide spatial and temporal range.

## Discussion

In summary, we introduce the EAT imaging platform: a non-invasive, label-free method demonstrating high voltage sensitivity and exceptional spatiotemporal resolution. The image contrast provided by EAT primarily originates from the distribution of

electric energy within tissues, thereby enabling the imaging of eletroration dynamics in soft tissues. The technique has an unprecedented dynamic monitoring range from tens to thousands of volts per centimeter and can image tissue at depths of over 7.5 cm (Supplementary Fig. S4, Supplementary Note. S4) with a spatial resolution of <0.2 mm and temporal resolution of 10 ms, with the potential for even higher resolutions. To clarify, the primary goal of EAT imaging in our research is not to directly visualize the distribution of the electric field or specific electrical conductivity. Rather, it's aimed at imaging the distribution of electrical energy deposition. This deposition is directly linked to

the electroporation treatment, and by mapping it, EAT imaging can yield valuable insights into the effectiveness and scope of electroporation. This platform can function under in vivo conditions, facilitating real-time observations during electroporation. Such capability could enhance the precision of electroporation treatment and holds immense potential in the field of biomedicine.

The electrical pulse duration is crucial in this study. Electroacoustic tomography is based on Joule heating-induced thermal expansion, which generates an electroacoustic signal[35]. The thermal diffusion and volume expansion during the pulse duration affect the image resolution[36]. EAT is suitable for nanosecond electroporation applications, but it can also be used for microsecond electroporation applications with relatively lower resolution. Real-time electric field intensity monitoring with high temporal resolution is still valuable.

Irreversible Electroporation (IRE) is a recognized non-radiative method for tumor ablation, but aggressive treatment planning can damage healthy tissue[42]. Thanks to the ultra-short pulse duration at the nanosecond level, this method is also considered a nonthermal treatment method. Understanding the spatial distribution of cell death during electroporation therapy is critical for precise treatment planning[43]. However, the biophysical mechanisms of IRE remain unclear, particularly in the complexity of the human body[44]. IRE pulse sequences can affect different tissues differently[45], making treatment planning challenging. Several studies have attempted to determine electric field thresholds for IRE in different tissues[46,47], but there is a complex relationship between cell survival and pulse width, number, spacing, intensity, and repetition frequency[14]. Real-time dosimetric tools may be necessary to determine the extent of IRE. EAT can reflect electric field and energy deposition in tissue in real-time, making it a potential guide for IRE treatment, but further analysis is needed to understand the relationship between EAT images and electric field energy distribution. This research utilizes uniform samples for conducting experiments, but it is crucial to acknowledge that imaging outcomes might differ when dealing with more complex scenarios involving diverse tissues. Additionally, differences in acoustic properties can introduce discrepancies in the transmission of the EA signal, which could potentially lead to amplified artifacts or distorted images during reconstruction. However, by incorporating ultrasound imaging and employing correction techniques to address the distortions caused by varying tissue acoustic properties, these challenges can be effectively mitigated. Advanced data processing and image reconstruction algorithms play a vital role in managing these issues successfully.

Like other radiation-induced acoustic imaging systems such as laser-induced photoacoustic imaging[36] and X-ray-induced acoustic imaging[41], EAT imaging seamlessly adapts to existing ultrasound systems. This compatibility reduces equipment and training costs, utilizing existing ultrasound imaging infrastructure, and thereby enhancing its potential for clinical application. Specifically, EAT can be conveniently adapted to clinical linear probes, facilitating the feasibility of US/EAT dual-modality imaging. This adaptability improves the precision of electroporation treatment. However, the limited viewing angle inherent to linear probes can distort EAT images[48]. To overcome this, strategies like deep learning and iterative reconstruction approaches can be employed[49–51]. Moreover, various algorithms can further enhance image resolution and frame generation rate[52–54]. In our future work, we plan to carry out postoperative histological analysis of irreversible electroporation (IRE) treatment and compare the results with EAT images. Moreover, electroporation shows potential as a radiation-free alternative to high-dose brachytherapy for prostate cancer treatment. This

suggests that the use of transrectal EAT/US hybrid imaging could be highly valuable for real-time monitoring of the ablation area. With the promising advancements in the development of handheld 2D planar ultrasound transducer arrays, the prospect of achieving in vivo three-dimensional (3D) imaging becomes feasible[55]. The adoption of 3D volumetric EAT imaging could offer physicians a more comprehensive real-time view of tissue information, further enhancing its potential clinical applications.

## Methods

**Setup**. The experimental setup for electroacoustic tomography consists of two parts (shown in Fig. 1b): (1) the circuit design of the electrical pulse generator and (2) the acoustic signal acquisition system. The electrical pulse generator is adjustable in pulse voltage (0 ~ 2 kV), pulse width (100 ns~1 ms), and pulse repetition frequency (0 ~ 1 MHz). A pair of tungsten electrodes (573400 & 57400, Parylene C-Insulated Tungsten Microelectrode, A&M System, USA) releases the electrical pulse to induce acoustic emission in the tissue, while an insulating material mostly encapsulates the electrodes, leaving only the tip (~1 mm) exposed. A 3D-printed fixture holds the electrodes in place with a spacing of 1 mm or 5 mm depending on experimental needs (Supplementary Fig. S1, Supplementary Note. S1). The signal of the electrical pulse is collected by an attenuation probe (P4250, Keysight Technologies, USA) and fed to a delay generator (DG535, STANFORD RESEARCH, USA) to obtain a TTL trigger signal that can be received by the data acquisition system.

We utilized two acoustic detection systems for the study: (1) a single-element probe-based system for EA signal acquisition and analysis, and (2) a ring-array-based parallel acquisition system for EAT image reconstruction. The non-focused immersion ultrasound transducer (AS309-S, Olympus, USA) of the single-element probe system was placed outside a custom acrylic water tank with an opening sealed by a polyethylene film for signal reception. The received signal was amplified by a custom preamplifier (62 dB, PhotoSound, USA) and fed to an oscilloscope (Keysight Technologies, USA) for data caching. The data were then transferred to a PC and analyzed using MatLab (2021b). For the ultrasound ring-array system, a 3D-printed tank was used to match a custom 128-channel ultrasound ring-array transducer (Doppler Ltd., Guangzhou, China), and the signals were digitized by a 128-channel data acquisition (SonixDAQ, Ultrasonics, BC, Canada) and then processed using Matlab (2021b).

**Signal processing and image formation**. The electroacoustic signal was denoised using wavelet-based methods, filtered based on its spectrum and averaged to remove noise. The wavelet filter used is based on DWT, using coif wavelets, and incorporates a sqtwolog threshold selection method to retain bandwidth limited signals[56]. The SNR of the data has improved from 17.8 dB to over 55 dB. Sound velocity mismatch was not considered due to the homogeneity and small size of the phantom. EAT image reconstruction was performed using a filtered back-projection algorithm[39] and Hilbert-transformed signals.

**Sample preparation**. Saline agar phantoms with 1% salt concentration were used in EAT characterization experiments. All materials were weighed with an electronic balance, and the agar salt solution was heated to 200 °C on a magnetic stirrer before being lowered to 105 °C when boiling began. The solution was stirred continuously until all visible bubbles disappeared, then poured into a 3D-printed mold with dimensions of 40 mm x 40 mm x 40 mm and allowed to cool and solidify. Freshly made

agar phantoms were used for each experiment to avoid salt precipitation when immersed in water.

Fresh chicken breasts were used as the sample material for in vitro experiments. The chicken breasts were cut into cylindrical shapes with a consistent thickness of 20 mm x 20 mm x 20 mm after being lightly frozen. Before the experiments, the chicken breast and electrodes were centered on the ring array and prefixed. To minimize the effects of osmotic pressure variations on the results, water was injected.

**Simulation on electrical field distribution**. Finite element methods (FEM) were employed to simulate electric field distributions using COMSOL Multiphysics® (COMSOL, v6.0, Stockholm, Sweden). A 2D domain was used, consisting of two circular tungsten electrodes within a square muscle tissue environment. Depending on experimental requirements, the two electrodes were either connected to positive and ground (1p1g) or both to positive (2p). The voltage applied to the electrodes was the same as that used in the experiment. Acoustic simulations were performed using the MatLab K-Wave toolbox, and the electric field energy deposition distributions obtained from the finite element simulations were used to calculate the initial sound pressure after interpolation. The ultrasonic transducer consisted of 128 elements distributed on a ring with a radius of 50 mm. Thermal noise with a random distribution was added to the EA signal detected by the ultrasound transducer. The EAT image was then reconstructed using a two-dimensional time-reversed image reconstruction algorithm.

**Reporting summary**. Further information on research design is available in the Nature Portfolio Reporting Summary linked to this article.

## Data availability

Sampled ultrasound waveforms are available through zenodo: https://doi.org/10.5281/zenodo.8132772. The authors declare that all other data supporting the findings of this study are available from the corresponding author upon reasonable request.

## Code availability

The back-projection recontruction code is available through zenodo: https://doi.org/10.5281/zenodo.8132772.

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

## Acknowledgements
This work was supported by UCI Chao Family Comprehensive Cancer Center (P30CA062203).

## Author contributions
Y.X. derived the analytical model, initiated the experiments, and completed the article writing. L.S. created the k-wave simulations and image reconstructions. S.W. provided advice on the experimental design. Y.Y. provided assistance during the experiments. P.P. contributed to the electroacoustic theory development. V.N. developed the nanoseconds pulsed electric field generator. L.X. conceived and supervised the project.

## Competing interests
The authors declare no competing interests.
