## [Peer Review File · Communications Engineering]

Reviewers' comments:

Reviewer #1 (Remarks to the Author):

The manuscript introduces electroacoustic tomography (EAT) as an innovative imaging modality for real-time monitoring of deep tissue electrical fields during electroporation treatment. The authors provide a comprehensive description of the EAT imaging system, the underlying principle of EAT, and experimental results obtained in water and soft tissue. The results highlight the remarkable capability of EAT to visualize the distribution of electrical fields during electroporation, offering high-resolution and real-time monitoring. The manuscript is well-written, the technology is highly innovative, and the study makes a significant contribution to addressing a critical challenge in the field. Therefore, it is highly recommended for publication in Communications Engineering. However, before publication, a few points should be addressed.

Specific Comments:

Introduction: The introduction provides a comprehensive background on the applications and importance of electroporation in biotechnology and medicine. However, it would be helpful to briefly mention the limitations of existing monitoring methods or why real-time monitoring during pulse delivery is challenging with current medical imaging modalities.

Methods: The methods section provides a detailed description of the experimental setup and data processing steps. However, some additional information is needed to clarify the following points:

a. In the setup section, it would be helpful to mention the specifications of the ultrasound transducer used in the single-element probe-based system, for example, how many dB has been used in the pre-amplifier system?

b. In the signal processing and image formation section, more details should be provided on the specific wavelet-based denoising method used and the parameters chosen for filtering and averaging the electroacoustic signal.

Results: The results section presents the experimental findings and includes figures demonstrating the capabilities of EAT imaging. The results are clear and support the claims made by the authors. However, it would be beneficial to include quantitative measurements or metrics to assess the performance of the EAT imaging system, such as signal-to-noise ratio (SNR) values or spatial resolution measurements.

Discussion: The discussion provides a good overview of the potential applications and benefits of EAT imaging in electroporation therapy. However, it would be valuable to discuss the limitations and challenges of EAT imaging, such as the impact of tissue heterogeneity on image quality.

Conclusion: The conclusion is well-written and summarizes the main findings of the study. However, it would be helpful to include a brief statement on the future directions or next steps for further development and validation of EAT imaging for electroporation monitoring in clinical settings.

Minor Comments:

1. It would be helpful to provide a legend for all the abbreviations used in the manuscript to aid readers in understanding the text.
2. In the caption of Figure 3, the phrase "The standard deviation of PA amplitude (error bars), and a linear fit of PA amplitude (solid line)" is confusing.
3. Please explain where the ring artifacts is coming from in Fig. 4(e). Why the EAT image reconstruction in Fig. 4(d) is so noisy?

4. The references cited in the manuscript should follow a consistent format (e.g., APA or IEEE style) throughout the text and in the reference list.

Overall, the study presents a novel approach using EAT imaging for real-time monitoring of deep tissue electroporation. The experimental results are promising and demonstrate the potential of EAT imaging in the field. Addressing the minor comments and providing additional clarification and quantitative measurements will enhance the manuscript and contribute to its scientific impact.

Reviewer #2 (Remarks to the Author):

The authors proposed electroacoustic tomography (EAT) to provide real-time assessment of deep tissue electroporation. The study is solid and detailed. However, there are still several technical questions that should be addressed. Please see the detailed comments below.

1. The authors should state more clearly the contrast of EAT. And how this contrast should be interpreted and how it could be useful for medical assessment.
2. The measured initial pressure is directly related to the electrical field distribution and specific electrical conductivity. However, different tissue may have different conductivity, which will generate a systematic bias for EAT measurements. This should be addressed.
3. The authors claimed a high resolution, however, which was not quantified in the manuscript. Quantification of spatial resolutions should be added to the manuscript.
4. Will the repeated electrical stimulation increase the temperature of the tissue? If so, the speed of sound will become another variable during reconstruction as it changes with temperature. This should be discussed in the manuscript.

Reviewer #3 (Remarks to the Author):

The article presents study on development of EAT imaging system and experimental validation studies using Agar and ex-vivo (chicken breast) samples. The proposed modality is of great importance to clinical applications. I have certain clarifications:

1. The proposed imaging modality is invasive in nature – where electrodes are inserted so as to induce (pulse) electric potential and thus transient heat. In the end clinical, where the subject of interest is human, how we can handle this technical concern.
2. The author mentions “EAT is capable of providing submillimeter resolution at depths of up to 7.5 centimeters,” It is required to include experimental data.
3. The authors mention that “The application of electricity in humans can be traced back to the 18th century when electric fields were first observed to cause tissue damage [1-4].” Please include the very reference.
4. It is mentioned that “Electroporation-based therapies offer new ways of delivering therapeutic agents into cells, with the efficacy of these treatments depending on the parameters of the electrical pulses (including amplitude, pulse duration, and the number of pulses).” Is it independent of pulse repetition

frequency?

5. Photoacoustic imaging (PAI) – which is not only non-invasive imaging modality but also provides vital molecular and patho-physiological parameters – has been emerging as a potential and promising candidate while addressing the mentioned challenges (more specifically in Introduction) in this article. A comparison (though qualitative) is required as the proposed imaging modality is more or less similar to PAI.

6. What is threshold safety limit of electric voltage to be delivered in human subject? Compare with the typical voltage employed in this study.

7. Please check the dimensionality of LHS and RHS of Eq. 2.

8. “and this allows for expressing the initial pressure rise in the target [33].” Is preferred to write as: “and this allows for expressing the initial pressure rise in the target as [33]:”

9. Here is some typo mistakes. Example, Fig. 2(d) and (e). Repeated words in “Filtering the electroacoustic signal with 6 MHz low pass filter results in an SNR of 28.5 dB.MHz low pass filter results in an SNR of 28.5 dB”.

10. With reference to Fig. 5, the initial frame corresponding to $t = 375\text{msec}$. gives no signals. Why is it so? What is the time-scale of generation of acoustic signals from electric pulse? In the case of photoacoustic – which is the generation of sound upon illumination by pulse light beam – it is instantaneous ($\sim\text{nsec}$) and lime-time of PA signals are of the order of $\sim\text{usec}$. PAI and the proposed modality are closely similar (see Eq. 1) where optical fluence is just replaced by the incident pulse electric parameter. Assuming acoustic speed $\sim 1500\text{m/sec}$ (which is 1.5m/msec), the time duration for acoustic signal to traverse $\sim\text{cm}$ is of just the fraction of msec.

11. Resolution can further improved by employing DMAS techniques reported.

- Delay-and-sum-to-delay-standard-deviation factor: a promising adaptive beamformer
- Higher-order correlation based real-time beamforming in photoacoustic imaging, J. Opt. Soc. Am. A 39(10), 1805-1814 (2022). <https://doi.org/10.1364/JOSAA.461323>
- Simplified-delay-multiply-and-sum based promising beamformer for real-time photoacoustic imaging, IEEE Trans. Instrum. Meas. 71, 4006509 (2022). <https://doi.org/10.1109/TIM.2022.3187734>

Response Letter

Manuscript Number: # COMMS-23-0167-T

Title: Electroacoustic Tomography for Real-time Visualization of Electrical Field Dynamics in Deep Tissue during Electroporation

Authors: Yifei Xu, Leshan Sun, Siqi Wang, Yuchen Yan, Prabodh Pandey, Vitalij Novickij, Liangzhong Xiang

1Department of Radiological Sciences, University of California at Irvine, Irvine, CA 92697

2The Department of Biomedical Engineering, University of California, Irvine, CA 92617

3Beckman Laser Institute & Medical Clinic, University of California, Irvine, Irvine, CA 92612

4Institute of High Magnetic Fields, Vilnius Gediminas Technical University, Vilnius, Lithuania

5Department of Immunology, State Research Institute Centre for Innovative Medicine, Santariškių 5, 08410, Vilnius, Lithuania

* co-corresponding author: liangzhx@uci.edu

* co-corresponding author: vitalij.novickij@vilniustech.lt

Article type: Original research

Correspondence to:

*Liangzhong (Shawn) Xiang, PhD
Associate Professor of Radiology & BME
University of California, Irvine
Affiliated faculty, Beckman Laser Institute and Medical Clinic
Affiliated faculty, Chao Family Comprehensive Cancer Center

825 Health Sciences Rd.
Medical Sciences, B-134
Irvine, CA 92697-5000
Email: liangzhx@hs.uci.edu
Tel: 949-8243544
Lab website: <https://truelab.som.uci.edu/>

Point-by-Point Response to Comments

Reviewer 1

Reviewer #1 overall comment:

“The manuscript introduces electroacoustic tomography (EAT) as an innovative imaging modality for real-time monitoring of deep tissue electrical fields during electroporation treatment. The authors provide a comprehensive description of the EAT imaging system, the underlying principle of EAT, and experimental results obtained in water and soft tissue. The results highlight the remarkable capability of EAT to visualize the distribution of electrical fields during electroporation, offering high-resolution and real-time monitoring. The manuscript is well-written, the technology is highly innovative, and the study makes a significant contribution to addressing a critical challenge in the field. Therefore, it is highly recommended for publication in Communications Engineering. However, before publication, a few points should be addressed.”

Response

We greatly appreciate your encouraging feedback. We have dedicated significant effort to revising the paper based on your valuable comments..

Comment #1

“Introduction: The introduction provides a comprehensive background on the applications and importance of electroporation in biotechnology and medicine. However, it would be helpful to briefly mention the limitations of existing monitoring methods or why real-time monitoring during pulse delivery is challenging with current medical imaging modalities.”

Response

Thank you for your suggestion. In response, we have incorporated additional details highlighting the limitations of current monitoring methods employed in electroporation treatment.

Revision

Line 42-47 “CT imaging is commonly used to guide the placement of electrodes during IRE ablation for liver cancer treatment³⁴, but it requires the patient to remain still during the procedure to prevent motion artifacts caused by patient breath and can expose them to higher radiation levels than ultrasound-guided IRE ^{35,36}. Both CT and MRI imaging can be used to assess the efficacy of IRE post-treatment, and contrast agents may be used in some cases to enhance visualization in cases where residual viable tumor tissue needs to be identified and further treatment is required^[29].”

Comment #2

“Methods: The methods section provides a detailed description of the experimental setup and data processing steps. However, some additional information is needed to clarify the following points:
a. In the setup section, it would be helpful to mention the specifications of the ultrasound transducer used in the single-element probe-based system, for example, how many dB has been used in the pre-amplifier system?
b. In the signal processing and image formation section, more details should be provided on the specific wavelet-based denoising method used and the parameters chosen for filtering and averaging the electroacoustic signal.”

Response

Thanks for your suggestion. We have incorporated more details into the Methods section of the manuscript. For point a, we used a customized preamplifier from *PhotoSound Technologies Inc*,

with a gain of 62dB. For point b, we employed a combination of a discrete wavelet transform (DWT)-based filtering method and a morphological expansion-based thresholding approach to retain bandwidth-limited signals. To determine the optimal hyperparameters, we conducted a grid search on a simulated dataset. The results indicated that using *coif5* wavelets, *sqtwolog* threshold selection, and a morphological window size of 9 yielded the best performance. It is noteworthy that WL filtering showed comparable or superior performance compared to LP filtering. We have added the specifics of the wavelet filter in the Signal processing and image formation section.

Revision

Line 273-275 "The received signal was amplified by a custom preamplifier (62dB, PhotoSound, USA) and fed to an oscilloscope (Keysight Technologies, USA) for data caching."

*Line 282-284 "The wavelet filter used is based on DWT, using *coif* wavelets, and incorporates a *sqtwolog* threshold selection method to retain bandwidth limited signals."*

Comment #3

"Results: The results section presents the experimental findings and includes figures demonstrating the capabilities of EAT imaging. The results are clear and support the claims made by the authors. However, it would be beneficial to include quantitative measurements or metrics to assess the performance of the EAT imaging system, such as signal-to-noise ratio (SNR) values or spatial resolution measurements."

Response

Thanks for pointing out this problem. The SNR information is added to the Signal Processing and image formation section. As for the spatial resolution, the object of EAT is the electric field. And for a continuous electric field, the limit of imaging resolution lies in the resolution of the imaging system. For our system, that limit lies in the acoustic part. Therefore, we cite the results of a previous resolution test for this acoustic acquisition system and consider the resolution to be 138 μm . A similar definition of resolution has been used in other prior papers. For example, Horng, J. et al. in their article "Imaging electric field dynamics with graphene optoelectronics" (Nat Commun 7, 13704 (2016)) used microspheres to measure their spatial resolution of the optical system and used it as a resolution for electric field measurements.

Revision

Line 284 "The SNR of the data has improved from 17.8 dB to over 55 dB."

Comment #4

"Discussion: The discussion provides a good overview of the potential applications and benefits of EAT imaging in electroporation therapy. However, it would be valuable to discuss the limitations and challenges of EAT imaging, such as the impact of tissue heterogeneity on image quality."

Response

We appreciate your important suggestion! More discussion on the limitations of the EAT has been added.

Revision

Line 239-248 "This research utilizes uniform samples for conducting experiments, but it is crucial to acknowledge that imaging outcomes might differ when dealing with more complex scenarios involving diverse tissues. Additionally, differences in acoustic properties can introduce discrepancies in the transmission of the EA signal, which could potentially lead to amplified artifacts or distorted images during reconstruction. However, by incorporating ultrasound imaging and employing correction techniques to address the distortions caused by varying tissue acoustic properties, these challenges can be effectively mitigated. Advanced data processing and image reconstruction algorithms play a vital role in managing these issues successfully."

Comment #5

“Conclusion: The conclusion is well-written and summarizes the main findings of the study. However, it would be helpful to include a brief statement on the future directions or next steps for further development and validation of EAT imaging for electroporation monitoring in clinical settings.”

Response

Thanks a lot for your suggestion. Based on your input, we have expanded our discussions to encompass further development in the clinical setting..

Revision

Line 246-254 “In our future studies, we intend to perform postoperative histological analysis of IRE (Irreversible Electroporation) treatment, comparing the obtained results with EAT images. Additionally, electroporation exhibits promise as a non-radiation alternative to high-dose brachytherapy in the treatment of prostate cancer. Considering this potential, the application of transrectal EAT/US hybrid imaging holds significant value for real-time monitoring of the ablation area.

Furthermore, there have been notable advancements in the development of handheld 2D planar ultrasound transducer arrays, showcasing encouraging progress. These advancements suggest the feasibility of achieving in vivo three-dimensional (3D) imaging [55]. By incorporating 3D volumetric EAT imaging, physicians can gain a more comprehensive and real-time understanding of tissue information, further augmenting its potential for clinical applications.”

Comment #6

“It would be helpful to provide a legend for all the abbreviations used in the manuscript to aid readers in understanding the text.”

Response

Thank you for bringing this up! The legend of the abbreviations has been added at the end of the figure description.

Revision

Line 343-345 “Figure 1..... PC: personal computer; TRIG: trigger signal; FG: function generator; nsEP: nanoseconds electrical pulse; PRE-AMP: pre-amplifier; DAQ: data acquisition system.”

Comment #7

“In the caption of Figure 3, the phrase “The standard deviation of PA amplitude (error bars), and a linear fit of PA amplitude (solid line)” is confusing.”

Response

Thanks for pointing out this. We have revised this sentence for better understanding.

Revision

Line 262-263 “The error bar (blue), and a curve fit of EA signal amplitude (red).”

Comment #8

“Please explain where the ring artifacts is coming from in Fig. 4(e). Why the EAT image reconstruction in Fig. 4(d) is so noisy?”

Response

Thank you for pointing out this issue. The EA signal is primarily generated around the electrode, with its highest intensity found near the electrode tip. As the ultrasound signal travels towards the opposing electrode, it encounters a reflection, resulting in the formation of a ring artifact.. Consequently, the radius of the ring artifact directly corresponds to the spacing between the electrodes. In the reconstructed image, only a single ring artifact is visible. This occurrence can be attributed to the minor variation in height among the handcrafted electrode tips, which permits only one electrode tip to generate a pronounced reflection of the ultrasound signal.. In the simulation results of Fig. 4(d), we artificially add noise to simulate the random noise that may be generated in the real situation, so the image appears quite noisy.

Comment #9

“The references cited in the manuscript should follow a consistent format (e.g., APA or IEEE style) throughout the text and in the reference list.”

Response

Thank you for your valuable suggestion. We will ensure that the citation format is corrected in the final draft that follows..

Reviewer 2

Reviewer #2 overall comment:

The authors proposed electroacoustic tomography (EAT) to provide real-time assessment of deep tissue electroporation. The study is solid and detailed. However, there are still several technical questions that should be addressed.

Response

Thank you very much for your positive feedback. We have carefully reviewed the comments and suggestions and made significant refinements to the paper accordingly.

Comment #1

“The authors should state more clearly the contrast of EAT. And how this contrast should be interpreted and how it could be useful for medical assessment.”

Response

Thank you for your suggestion. Equation (2) establishes a relationship between the initial acoustic pressure and various factors, including the square of the electric field ($E(r)^2$), electrical conductivity ($\sigma(r)$), thermal coefficient ($\beta(r)$), isothermal compressibility ($\kappa(r)$), mass density ($\rho(r,t)$), and specific heat capacity ($C_v(r)$). Based on this equation, we can infer that EAT imaging has the potential to provide valuable insights in two key areas: 1) Mapping the distribution of electrical field energy deposition during electroporation delivery: By visualizing the spatial distribution of the electric field energy, EAT imaging can shed light on how the energy is distributed within the target volume. 2) Assessing the dose amount deposited in the target volume: EAT imaging can provide information regarding the quantity of energy deposited in the target volume, enabling an evaluation of the efficacy of the electroporation treatment. These pieces of information are crucial in determining the effectiveness of electroporation as a treatment modality. We have also included a discussion on the EAT image contrast in the revised manuscript.

Revision

Line 69-76 “where $\beta(r)$ is the thermal coefficient (K-1), $\kappa(r)$ denotes the isothermal compressibility (Pa-1), $\rho(r,t)$ is the mass density (g m-3), $C_v(r)$ is the specific heat capacity at constant volume (J g-1 K-1), and $H(r,t)$ denotes the deposited electrical power density (J m-3) in tissue. The electroacoustic wave's amplitude correlates to the energy deposited, allowing for the reconstruction of electric field distribution. Joule's and Ohm's laws determine the relationship between electrical power density and electric field, and this allows for expressing the initial pressure rise in the target as ... where $E(r)$ is the electric field (V m-1) at the position of r and $g(t)$ denotes the voltage pulse width (s). $\sigma(r)$ denotes the specific electrical conductivity (S m-1), while it is not a constant.”

Line 202-205 “In summary, we introduce the EAT imaging platform: a non-invasive, label-free method demonstrating high voltage sensitivity and exceptional spatio-temporal resolution. The image contrast provided by EAT primarily originates from the distribution of electric energy within tissues, thereby enabling the imaging of electroporation dynamics in soft tissues.”

Comment #2

“The measured initial pressure is directly related to the electrical field distribution and specific electrical conductivity. However, different tissue may have different conductivity, which will generate a systematic bias for EAT measurements. This should be addressed.”

Response

We agree and we understand that the initial acoustic pressure is not solely determined by the electrical field distribution or specific electrical conductivity individually. It is indeed influenced

by the electrical energy distribution, which is the product of the electrical field distribution and specific electrical conductivity.

To clarify, the purpose of EAT imaging in our research is not to directly image the electrical field distribution or specific electrical conductivity. Instead, it aims to image the distribution of electrical energy deposition. This electrical energy deposition is directly associated with the electroporation treatment, and by visualizing it, EAT imaging can provide valuable information about the effectiveness and impact of electroporation. We have added a discussion in the revision.

Revision

Line 202-207 “To clarify, the purpose of EAT imaging in our research is not to directly image the electrical field distribution or specific electrical conductivity. Instead, it aims to image the distribution of electrical energy deposition. This electrical energy deposition is directly associated with the electroporation treatment, and by visualizing it, EAT imaging can provide valuable information about the effectiveness and impact of electroporation”

Comment #3

“The authors claimed a high resolution, however, which was not quantified in the manuscript. Quantification of spatial resolutions should be added to the manuscript.”

Response

Thank you for your suggestion. The quantification of the spatial resolution of the system we use is introduced in the section *Real-time visualizing electrical field dynamics* section. “The spatiotemporal dynamics of the electric field are captured at a temporal resolution of 10 ms. And the spatial resolution of 138 μ m has been demonstrated in this system.”

Sorry for missing this important information. The ultrasound ring array has been used in our previous publication [40], the imaging resolution of the ultrasound ring array has been thoroughly quantified and measured to be 138 μ m. We have taken note of this critical detail and have included it in the revised manuscript. Thank you for drawing our attention to this matter.

Comment #4

Will the repeated electrical stimulation increase the temperature of the tissue? If so, the speed of sound will become another variable during reconstruction as it changes with temperature. This should be discussed in the manuscript.

Response

Thank you for your insightful suggestion. We have taken into consideration the ultra-short pulse duration and low repetition frequency utilized in our experiments, which have resulted in an insignificant increase in tissue temperature. This important point has been incorporated into the discussion section of the manuscript..

Revision

Line 217-218 “Thanks to the ultra-short pulse duration at the nanosecond level, this method is also considered a non-thermal treatment method.”

Reviewer 3

Reviewer #3 overall comment:

The article presents study on development of EAT imaging system and experimental validation studies using Agar and ex-vivo (chicken breast) samples. The proposed modality is of great importance to clinical applications.

Response

Thank you for your positive feedback regarding our work on Electroacoustic Tomography. We greatly appreciate your kind suggestions, and we have made significant refinements to the article based on your valuable input.

Comment #1

“The proposed imaging modality is invasive in nature – where electrodes are inserted so as to induce (pulse) electric potential and thus transient heat. In the end clinical, where the subject of interest is human, how we can handle this technical concern.”

Response

Thank you for highlighting this point. We agree that Irreversible Electroporation (IRE), is indeed a minimally invasive treatment procedure used for the ablation of tumors or lesions in various organs and it has been used in clinics. The primary goal of IRE is to selectively ablate targeted tissues while minimizing damage to surrounding healthy structures. This makes it particularly suitable for treating tumors or lesions in critical areas where preserving organ functionality and structure is crucial, such as the liver, pancreas, prostate, and soft tissues.

It is important to clarify that the focus of our work is not to develop a new type of imaging modality for noninvasive diagnostic imaging. Instead, our research is centered around the development of a novel imaging technology for image-guided interventions. This technology aims to provide real-time visualization and guidance during IRE procedures, enhancing the accuracy and efficacy of the treatment.

Comment #2

“The author mentions “EAT is capable of providing submillimeter resolution at depths of up to 7.5 centimeters....” It is required to include experimental data.”

Response

Thank you for bringing this issue to our attention. We apologize for the lack of clarity in labeling the data presented in Figure 4 of the supplemental material in the article. We have taken note of this and have made revisions to ensure that this section is now clearly labeled for improved understanding.

Revision

Line 119-201 “The technique has an unprecedented dynamic monitoring range from tens to thousands of volts per centimeter and can image tissue at depths of over 7.5 cm (Supplementary Data Fig. S4) with a spatial resolution of ...”

Comment #3

“The authors mention that “The application of electricity in humans can be traced back to the 18th century when electric fields were first observed to cause tissue damage [1-4].” Please include the very reference.”

Response

Thanks for your suggestion. We have introduced another reference directly used to explain this sentence.

Comment #4

“It is mentioned that “Electroporation-based therapies offer new ways of delivering therapeutic agents into cells, with the efficacy of these treatments depending on the parameters of the electrical pulses (including amplitude, pulse duration, and the number of pulses).” Is it independent of pulse repetition frequency?”

Response

The repetition frequency of the pulses is also an important influence in electroporation therapy. Thank you for pointing out this oversight, and we have revised it in the article.

Revision

Line 22-25 “Electroporation-based therapies offer new ways of delivering therapeutic agents into cells, with the efficacy of these treatments depending on the parameters of the electrical pulses (including amplitude, pulse duration, the number and the repetition rate of pulses)”

Comment #5

“Photoacoustic imaging (PAI) – which is not only non-invasive imaging modality but also provides vital molecular and patho-physiological parameters – has been emerging as a potential and promising candidate while addressing the mentioned challenges (more specifically in Introduction) in this article. A comparison (though qualitative) is required as the proposed imaging modality is more or less similar to PAI.”

Response

We acknowledge that Electroacoustic Tomography (EAT) and laser-induced photoacoustic imaging share a common aspect in terms of utilizing energy (laser or electrical) to generate acoustic waves. However, this is likely the only similarity between these two modalities, as they differ significantly in their underlying principles and applications.

Laser-induced photoacoustic imaging primarily relies on optical absorption contrast, which is frequently utilized for imaging blood vasculature and other targets with strong optical absorption properties. It utilizes laser energy to excite tissue, and the resulting acoustic waves are detected to create images based on optical absorption.

On the other hand, EAT captures acoustic signals directly related to tissue conductivity and the applied electrical field. Its objective is not to develop a new noninvasive diagnostic imaging modality, but rather to advance the field of image-guided interventions. EAT focuses on the development of a novel imaging technology specifically tailored for real-time visualization and guidance during Irreversible Electroporation (IRE) procedures. The aim is to enhance the accuracy and efficacy of these interventions. We have added a discussion in the revised manuscript.

Revision

Line 237-240: “ Like other radiation-induced acoustic imaging systems such as laser-induced photoacoustic imaging and X-ray induced acoustic imaging, EAT imaging seamlessly adapts to existing ultrasound systems. This compatibility reduces equipment and training costs, utilizing existing ultrasound imaging infrastructure, and thereby enhancing its potential for clinical application. . ”

Comment #6

“What is threshold safety limit of electric voltage to be delivered in human subject? Compare with the typical voltage employed in this study.”

Response

Thank you for your suggestion. We added the voltage used in the latest research on nanosecond electroporation in humans for comparison.

Revision

Line 151-154 “A previous study has successfully conducted nanosecond electroporation on humans using ultra-high voltages up to 30 kV, and its safety has been established. Therefore, this experiment was conducted well within the established safety threshold.”

Comment #7

“Please check the dimensionality of LHS and RHS of Eq. 2.”

Response

Thanks for your suggestion. We checked the equation and do find it should be explained more clearly. The heating function $H(r) = J(r)*E(r)$ according to Joule’s law, and $J(r) = \sigma(r)*E(r)$ according to Ohm’s law. Thus, the dimensions of RHS and LHS are equal. The detailed verification process is shown below:

$$p_0(r, t) = \frac{\beta(r) \sigma(r)}{\kappa(r) \rho(r, t) C_v(r)} E(r)^2 g(t)$$

dimensionality for LHS is: Pa

dimensionality for RHS is:

$$\begin{aligned} & \frac{\frac{1}{K} \frac{S}{m}}{\frac{1}{Pa} \frac{g}{m^3} \frac{J}{gK}} \times \frac{V^2}{m^2} \times sec \\ &= \frac{S \times Pa \times V^2 \times sec}{J} \\ &= \frac{\frac{A}{V} \times V^2 \times sec}{W \times sec} \times Pa \\ &= \frac{V \times A \times sec}{V \times A \times sec} \times Pa = Pa = LHS \end{aligned}$$

Revision

Line 68-70 “where $\beta(r)$ is the thermal coefficient (K-1), $\kappa(r)$ denotes the isothermal compressibility (Pa-1), $\rho(r, t)$ is the mass density (g m-3), $C_v(r)$ is the specific heat capacity at constant volume (J g-1 K-1), and $H(r, t)$ denotes the deposited electrical power density (J m-3) in tissue.”

Line 74-75 “where $E(r)$ is the electric field (V m-1) at the position of r and $g(t)$ denotes the voltage pulse width (s). $\sigma(r)$ denotes the specific electrical conductivity (S m-1), while it is not a constant.”

Comment #8

“and this allows for expressing the initial pressure rise in the target [33].” Is preferred to write as: “and this allows for expressing the initial pressure rise in the target as [33]:”

Response

Thank you for your suggestion, we have modified this sentence according to your suggestion.

Revision

Line 70-73 “The electroacoustic wave's amplitude correlates to the energy deposited, allowing for the reconstruction of electric field distribution. Joule's and Ohm's laws determine the relationship between electrical power density and electric field, and this allows for expressing the initial pressure rise in the target as [35]”

Comment #9

“Here is some typo mistakes. Example, Fig. 2(d) and (e). Repeated words in “Filtering the electroacoustic signal with 6 MHz low pass filter results in an SNR of 28.5 dB.MHz low pass filter results in an SNR of 28.5 dB”.”

Response

Thanks a lot for pointing this out, we have fixed this error.

Revision

Line 352-353 “(e) Filtering the electroacoustic signal with 6 MHz low pass filter results in an SNR of 28.5 dB.”

Comment #10

With reference to Fig. 5, the initial frame corresponding to $t = 375\text{msec}$. gives no signals. Why is it so? What is the time-scale of generation of acoustic signals from electric pulse? In the case of photoacoustic – which is the generation of sound upon illumination by pulse light beam – it is instantaneous ($\sim\text{nsec}$) and lime-time of PA signals are of the order of $\sim\text{usec}$. PAI and the proposed modality are closely similar (see Eq. 1) where optical fluence is just replaced by the incident pulse electric parameter. Assuming acoustic speed $\sim 1500\text{m/sec}$ (which is 1.5m/msec), the time duration for acoustic signal to traverse $\sim\text{cm}$ is of just the fraction of msec.

Response

Thanks for pointing this out. There are two reasons why no signal is displayed in the initial frame: (1) the voltage is only 100V at this point, which is at a low level; (2) the signal in this frame is below the threshold we want to display for the purpose of unifying the color bar and suppressing background noise.

Comment #11

Resolution can further improved by employing DMAS techniques reported.

- Delay-and-sum-to-delay-standard-deviation factor: a promising adaptive beamformer
- Higher-order correlation based real-time beamforming in photoacoustic imaging, J. Opt. Soc. Am. A 39(10), 1805-1814 (2022). <https://doi.org/10.1364/JOSAA.461323>
- Simplified-delay-multiply-and-sum based promising beamformer for real-time photoacoustic imaging, IEEE Trans. Instrum. Meas. 71, 4006509 (2022). <https://doi.org/10.1109/TIM.2022.3187734>

Response

Thanks for your suggestion, we have added a discussion on methods to further improve the resolution.

Revision

Line 245-246 “In addition, there are algorithms that can be used to further improve the resolution and frame generation rate of images [52-54].”

REVIEWERS' COMMENTS:

Reviewer #1 (Remarks to the Author):

The authors have well addressed the concerns of the reviewers, and I recommend its publication.

Reviewer #2 (Remarks to the Author):

The authors addressed all my comments. The manuscript can be accepted as is.

Reviewer #3 (Remarks to the Author):

My concerns are duely responded. I have no more clarifications. Preferably, it will be great if they are incorporated in the manuscript for better clarity to the audiences (not necessarily in the domain of EAT).